

# Light dependent courtship behavior in *Drosophila simulans* and *D. melanogaster*

Michael P. Shahandeh,  Cameryn Brock and  Thomas L. Turner

Ecology, Evolution, and Marine Biology, University of California, Santa Barbara, Santa Barbara, CA, United States of America

## ABSTRACT

Differences in courtship signals and perception are well-known among *Drosophila* species. One such described difference is the dependency on light, and thus presumably vision, for copulation success. Many studies have described a difference in light-dependent copulation success between *D. melanogaster* and *D. simulans*, identifying *D. simulans* as a light-dependent species, and *D. melanogaster* as a light-independent one. However, many of these studies use assays of varying design and few strains to represent the entire species. Here, we attempt to better characterize this purported difference using 11 strains of each species, paired by collection location, in behavioral assays conducted at two different exposure times. We show that, while there is a species-wide difference in magnitude of light-dependent copulation success, *D. melanogaster* copulation success is, on average, still impaired in the dark at both exposure times we measured. Additionally, there is significant variation in strain-specific ability to copulate in the dark in both species across two different exposure times. We find that this variation correlates strongly with longitude in *D. melanogaster*, but not in *D. simulans*. We hypothesize that differences in species history and demography may explain behavioral variation. Finally, we use courtship assays to show that light-dependent copulation success in one *D. simulans* strain is driven in part by both males and females. We discuss potential differences in courtship signals and/or signal importance between these species and potential for further comparative studies for functional characterization.

## INTRODUCTION

Courtship in *Drosophila* is a multimodal form of communication, involving chemosensory, auditory, tactile, and visual signals (*Greenspan & Ferveur, 2000*). Often, male signals are more conspicuous and easily observed, and thus are more widely studied; for example, males of many species produce a courtship song by extending and vibrating thier wings (*Spieth, 1952*). The resulting song can be recorded and separated into discrete parts, such as pulse and sine song (*Von Schilcher, 1976*), and quantified using metrics like inter-pulse interval and pulse duration (*Kyriacou & Hall, 1982*). These metrics show clear signs of species specificity that are important to conspecific reproductive success (*Kyriacou & Hall, 1982*; *Spieth, 1974*). Another signal that can be easily quantified, and has been extensively studied, is variation in chemotactic pheromones (*Cobb & Jallon, 1990*; *Jallon & David, 1987*; *Pardy et al., 2019*; *Pischedda et al., 2014*). These pheromones, present on the fly

Corresponding author
Michael P. Shahandeh,
mshahandeh@ucsb.edu

cuticle, sometimes act as sex- and species-specific identifiers that stimulate courtship among conspecific pairings, but suppress courtship between heterospecific pairs in some species (*Billeter et al., 2009*; *Manning, 1959b*; *Savarit et al., 1999*; *Shahandeh, Pischedda & Turner, 2018*; *Shahandeh & Thomas, 2020*).

In the case of visual signals, for *Drosophila melanogaster*, visual perception of a moving courtship target is necessary for males to initiate and maintain courtship (*Agrawal, Safarik & Dickinson, 2014*; *Cook, 1980*), with males preferring to initiate courtship towards moving targets over stationary ones (*Tompkins et al., 1982*). *D. melanogaster* males prefer larger females (*Byrne & Rice, 2006*; *Edward & Chapman, 2012*), though it remains to be determined if this choice is driven by visual perception or some other cryptic correlate of female quality. Given that these prior results highlight the potential importance of visual signaling in courtship, it is noteworthy that *D. melanogaster* is said to copulate successfully independent of light (*Manning, 1959a*; *Spieth & Hsu, 1950*). Indeed, for many other *Drosophila* species, copulation success is relatively light dependent (*Ewing, 1983*; *Grossfield, 1971*; *Spieth, 1974*). However, the specific visual signals that make courtship light dependent for other species remain relatively unclear, with a few notable exceptions where males or females have evolved an additional postural display whereby they presumably send visual signals using specific, repeated, positions or movements (*Brown, 1965*; *Ewing, 1983*).

For the commonly studied cosmopolitan species, *D. simulans*, no such postural behavior has been described. Nonetheless, this species is said to differ largely from *D. melanogaster* in their light-dependent copulation success (*Grossfield, 1971*; *Manning, 1959a*). Specifically, *D. melanogaster* copulates successfully independent of light, while *D. simulans* copulates significantly less in the dark (*Manning, 1959a*; *Spieth & Hsu, 1950*). The ubiquity of this purported difference is debatable, however, as genetically blinded *D. melanogaster* do not copulate as successfully as wild-type males when kept in bright light (*Tompkins et al., 1982*). Reported differences in *Drosophila* light-dependent copulation behavior may be a result of strain-specific behavior or may reflect experimentally induced variation. Indeed, studies often use just one or two strains as a representative of a species and conduct assays of variable lengths, ranging from minutes to a week, and designs, ranging from individual pairs of flies to large groups that are blinded or compared under varying light regimes (*Cobb & Ferveur, 1995*; *Giglio & Dyer, 2013*; *Gleason et al., 2012*; *Grossfield, 1971*; *Manning, 1959a*; *Spieth & Hsu, 1950*; *Tompkins et al., 1982*).

In the present study, we seek to more accurately quantify the level of light dependency for these two sister species of *Drosophila*: *D. melanogaster* and *D. simulans*. To do so, we measure light dependent copulation success at two exposure times for 11 strains of *D. melanogaster* and 11 strains of *D. simulans* collected from paired locations around the globe. By doing so, we are able to quantify species differences in light dependent copulation behavior as well as assess intraspecific variation and time-dependency. Further, there is some evidence that light-dependent copulation success is inversely correlated with ecological generality (*Grossfield, 1971*). Using our *D. melanogaster* and *D. simulans* strains, we also test for correlations among behavior and geographic variables to gain insight into potential factors underlying global behavioral variation. Finally, we use the most extreme

**Table 1** ***Drosophila simulans* and *D. melanogaster* strains.** For each strain we used, the National *Drosophila* Species Stock Center number is provided. Also provided are the collection location and date (when available), along with the longitude and latitude used to test for correlations of behavior and geography. For strains where a longitude and latitude was not provided by the stock center, we used the longitude and latitude for the collection location.

| Stock # | Species | Collection location (date) | Longitude | Latitude | Strain label |
|---|---|---|---|---|---|
| 14021-0251.005 | *D. simulans* | Lima, Peru (1956) | −77.0428 | −12.0464 | PER005 |
| 14021-0231.01 | *D. melanogaster* | Ica, Peru (1956) | −75.7342 | −14.0755 | M-PER01 |
| 14021-0251.009 | *D. simulans* | Gorak, New Guinea (1961) | 145.3863 | −6.0835 | NG009 |
| 14021-0231.120 | *D. melanogaster* | Port Moresby, Papua New Guinea (1982) | 147.1803 | −9.4438 | M-NG120 |
| 14021-0251.169 | *D. simulans* | South Africa | 22.9375 | −30.5595 | SA169 |
| 14021-0231.51 14021-0231.62 | *D. melanogaster* | Cape Town, South Africa (2007) | 18.4241 | −33.9249 | M-SA51 |
| 14021-0251.181 | *D. simulans* | Crete Island, Greece (2002) | 24.8093 | 35.2401 | GRE181 |
| 14021-0231.69 | *D. melanogaster* | Athens, Greece (1965) | 23.7275 | 37.9838 | M-GRE69 |
| 14021-0251.261 | *D. simulans* | Lujeri, Malawi (2009) | 35.6484 | −16.0400 | MAL261 |
| 14021-0231.76 | *D. melanogaster* | Lujeri, Malawi (2009) | 35.6484 | −16.04 | M-MAL76 |
| 14021-0251.288 | *D. simulans* | Athens, Georgia (2009) | 83.3576 | 33.9519 | GEO288 |
| 14021-0231.183 | *D. melanogaster* | Athens, Georgia (2009) | 83.3576 | 33.9519 | M-GEO181 |
| 14021-0251.004 | *D. simulans* | Australia (1955) | 133.7751 | −25.2744 | AUS004 |
| 14021-0231.03 | *D. melanogaster* | Queensland, Australia | 142.7028 | −20.9176 | M-AUS05 |
| 14021-0251.166 | *D. simulans* | IslaMorada, Florida | −80.6278 | 24.9243 | FLO166 |
| 14021-0231.14 | *D. melanogaster* | Orlando, Florida | −81.3792 | 28.5383 | M-FLO14 |
| 14021-0251.001 | *D. simulans* | Georgetown, Guyana (1956) | −58.1551 | 6.8013 | GUY001 |
| 14021-0231.15 | *D. melanogaster* | Bahia, Brazil | −41.7007 | −12.5797 | M-BRAZ15 |
| 14021-0251.196 | *D. simulans* | Ansirabe, Madagascar (1998) | 47.0291 | −19.873 | MAD196 |
| 14021-0231.125 | *D. melanogaster* | Tananarive, Madagascar (1982) | 47.5079 | −18.8792 | M-MAD125 |
| 14021-0251.006 | *D. simulans* | Nueva, California (1961) | −117.1459 | 33.8014 | NUE006 |
| 14021-0231.131 | *D. melanogaster* | La Jolla, California (2009) | −117.2713 | 32.8328 | M-SD131 |

lines from either side of the behavioral spectrum for each species in courtship observation to begin to understand the mechanistic causes of light-dependent behavior.

## MATERIAL AND METHODS

### Fly strains and maintenance

We selected 11 wild-type *D. melanogaster* and 11 wild-type *D. simulans* strains (Table 1) from the National *Drosophila* Species Stock center that were collected across 6 continents. These 22 strains constitute 11 pairs of *D. simulans* and *D. melanogaster* strains that were collected at approximately equal latitudes and longitudes. Whenever possible, we chose strains that were collected from the same location at the same time.

We maintained each strain on non-overlapping, alternating 2-week life cycles. We reared all strains on a standard cornmeal-yeast-molasses medium in 25 mm vials at 25 °C and ∼50% humidity under a 12:12 h light/dark cycle. At the beginning of each cycle, we transferred roughly 20-30 adult flies to a culture vial with fresh media. We allowed the flies to oviposit for 48 h before transferring them to a second collection vial with fresh media, where flies were allowed to oviposit for an additional 24 h before being discarded. We

repeated this process every fourteen days using offspring from the culture vials to maintain each strain. For use in experiments, we collected male and female offspring as virgins from the collection vials 4–5 h following "lights-on" under light $CO_2$ anesthesia 11 days after oviposition. For all experiments described below, we aged males and females separately in holding vials with fresh food media at a density of 10 flies for 3–4 days before each assay. We aged flies in groups prior to assay, because flies held in isolation display increased aggressive behaviors we were concerned would affect copulation success, skewing our data (*Hoffmann, 1990*).

### Light dependent copulation success assay

To measure each strain's ability to successfully copulate independent of light, we measured copulation success in a normal light (control) treatment, and in an entirely dark (experimental) treatment, side-by-side on the same day. On the morning of each assay, immediately following "lights-on", we aspirated a single virgin male and female into a 20 mm vial with fresh food media sealed with a foam plug. We chose to use single pairs, as there is some evidence from other *Drosophila* that males approach females sequentially in the wild, and females that are approached singly are more likely to copulate (*Noor & Ortiz-Barrientos, 2006*). We assayed flies in vials with food media because adult *Drosophila* are most likely to encounter mates on or near a food substrate (*Soto-Yéber et al., 2018*). We held these vials at 25 °C and ~50% humidity for either 2 or 6 h in an incubator illuminated (control), with a Phillips "Cool white" 32-watt fluorescent light bulb or in the same incubator sealed in a light-proof box (experimental). We chose these time-points, rather than a day or week-long assay, because they represent a short, more realistic exposure time for flies in the wild, and a longer exposure time with which we could assess time dependency. At the end of the assay, we used an aspirator to remove the male from the vial so no post-assay copulation could occur. We then held females in vials at 25 °C and ~50% humidity in a 12 h:12 h light/dark cycle to oviposit for 7 days. On day seven, we checked each vial for the presence of larvae or early stage pupae, indicating whether insemination successfully occurred during the assay time. We collected all data on a weekly basis over the course of 6 weeks.

For each of the 11 *D. melanogaster* and *D. simulans* strains, we observed 25–31 pairings in the control and experimental treatments for each exposure time (2 or 6 h). For each strain, we calculated copulation success using the proportion of vials that produced offspring as a proxy for the proportion of vials where successful copulation occurred. We used Fisher's exact tests to compare copulation success between control and experimental treatments for each strain at each exposure time, followed by post-hoc correction for multiple comparisons (*Holm, 1979*). We tested for a species-wide difference between light/dark treatments at each exposure time using paired t-tests or paired Wilcoxon rank-sum tests when the data did not fit a normal distribution.

We next calculated relative dark copulation success for each exposure time as the percent of successful copulations in the dark treatment divided by the proportion of successful copulations for the same strain in the light treatment (% dark/% light). We compared relative dark copulation success between the 2-hour and 6-hour exposure times within

species, and for each exposure time between species, using paired t-tests. We also tested for a correlation between relative dark copulation success at each exposure time and three collection variables: longitude, the absolute value of latitude (i.e., distance from the equator), and collection date (when available). We used Pearson's correlation test when the data fit a normal distribution, and Spearman's rank correlation when the data was not normally distributed.

## Courtship assays

To test if males from *D. melanogaster* and *D. simulans* strains with relatively light-dependent and light-independent copulation success actively court females in the dark, we conducted courtship assays under two treatments. For each treatment, we gently aspirated single virgin males into vials containing a thin layer of food media 24 h prior to the assay. The morning of the assay, 1–2 h following "lights on," we aspirated a single female into the vial and pushed a foam plug down into the vial until it was just 1–2 cm from the food surface. Because these flies were held in a much smaller space, they were forced to interact even when held in the dark, so we observed courtship for just 30 min to avoid observer fatigue. We scored each minute for one of three easily distinguished courtship behaviors: (1) singing (single-wing extension and vibration), (2) attempted copulation, and (3) successful copulation. For males exhibiting multiple behaviors within a minute, we scored each pairing once per minute per behavior. As a control, we observed male and female pairings under bright light. As an experimental treatment, we observed male and female pairings in a dark room illuminated solely with red light because the *Drosophila* compound eye is insensitive to red wavelengths of light (*Salcedo et al., 1999*).

We selected the single most light-independent and light-dependent *D. simulans* strains (MAL261 and SA169, respectively) and *D. melanogaster* strains (M-BAZ15 and M-NG120, respectively) from the two-hour exposure period for courtship observation. First, for all four strains, we observed males with females of their own strain. For the *D. simulans* strains, we also observed males with females collected from the opposing strain because SA169 displayed significantly less frequent courtship towards SA169 females when observed in the dark (see results). In either treatment, we considered any male that spent more than 10% of the assay time exhibiting any courtship behavior as successfully courting. We compared the proportion of males that courted females in the light and in the dark using Fisher's exact tests followed with post-hoc correction for multiple comparisons (*Holm, 1979*). For males that displayed courtship, we also calculated courtship latency (the time from the start of assay until the male initiates courtship) and courtship effort (the total proportion of time a male spends courting during the 30-minute assay). If a pair successfully copulated, we calculated courtship effort as the percent of time a male spent courting from start of assay until copulation. We did not apply the same 10% cut-off for courtship latency and effort as we did for the proportion of males that courted, due to exceedingly low sample sizes. We compared courtship latencies and courtship efforts between pairings using Wilcoxon rank-sum tests followed by post-hoc correction for multiple comparisons (*Holm, 1979*).

## RESULTS

### Light dependent copulation success
#### D. simulans

For *D. simulans*, copulation is much more successful in the light than in the dark when males and females were held together for 2 h (paired $t$-test, $p < 0.0001$, Fig. 1A). On average, *D. simulans* strains were 41.77% as successful at copulating in the dark compared to in the light when given 2 h. Each of the 11 strains had decreased copulation success in the dark, and 8 of 11 strains were statistically significant after correcting for multiple comparisons (Fig. 1A, Table S1). The species-level pattern is still detectable when males and females are held together for 6 h, but far less significant (paired Wilcoxon, $p < 0.05$, Fig. 1B). Overall, *D. simulans* strains were 77.36% as successful at copulating in the dark as they were when held in the light for 6 h. This marked improvement appears to be driven by six lines (NUE006, MAL261, GUY001, FLO166, PER005, and MAD196), which copulate approximately equally as successfully in the light as they do in the dark when given increased time (all $p = 1$, Fig. 1B, Table S1). Still, the remaining lines displayed reduced copulation successes in the dark compared to the light treatment, with 3 remaining significantly lower following correction for multiple comparisons (Fig. 1B, Table S1). When we compared the relative dark copulation success for our *D. simulans* strains across the two exposure times, we found a significant difference between the 2 and 6-hour treatments (paired Wilcoxon $p < 0.001$), with strains showing decreased light dependency for copulation at the 6-hour exposure time (Fig. 1C). Overall, the proportion of flies that successfully copulated increased by 0.36, on average, when given extra time.

#### D. melanogaster

We also found that copulation is more successful in the light for *D. melanogaster* when pairs were given 2 h to mate (paired $t$-test $p < 0.001$, Fig. 2A). On average, *D. melanogaster* was 65.54% as effective at mating in the dark as they were in the light when given 2 h, with 6 of 11 strains significantly worse in the dark after correcting for multiple comparisons (Fig. 2A, Table S1). Like with *D. simulans*, we were still able to detect an overall effect of light vs. dark treatments at 6 h, albeit less significantly (paired Wilcoxon $p < 0.01$, Fig. 2B, Table S1). While just two lines individually copulated significantly less at this exposure time (M-MAD125 and M-NG120), all lines showed a reduced proportion of copulating pairs in the dark relative to the light. Overall, the difference between treatments at 6 h was smaller, with strains copulating 80.24% as successfully in the dark as they did in the light. Again, we find that *D. melanogaster* males show reduced light-dependent copulation behavior at 6 h relative to 2 h (paired Wilcoxon $p < 0.01$, Fig. 2C). Overall, the proportion of flies that successfully copulated increased by 0.15, on average, when given extra time.

### Comparing *D. simulans* and *D. melanogaster*

When we compare relative copulation success in the dark between *D. simulans* and *D. melanogaster*, we see a significant difference at the 2-hour exposure time (paired $t$-test $p < 0.05$). Specifically, the relative copulation success in the dark for *D. simulans* (41.77%) is significantly lower than that of *D. melanogaster* (65.54%). Contrastingly, we do not find

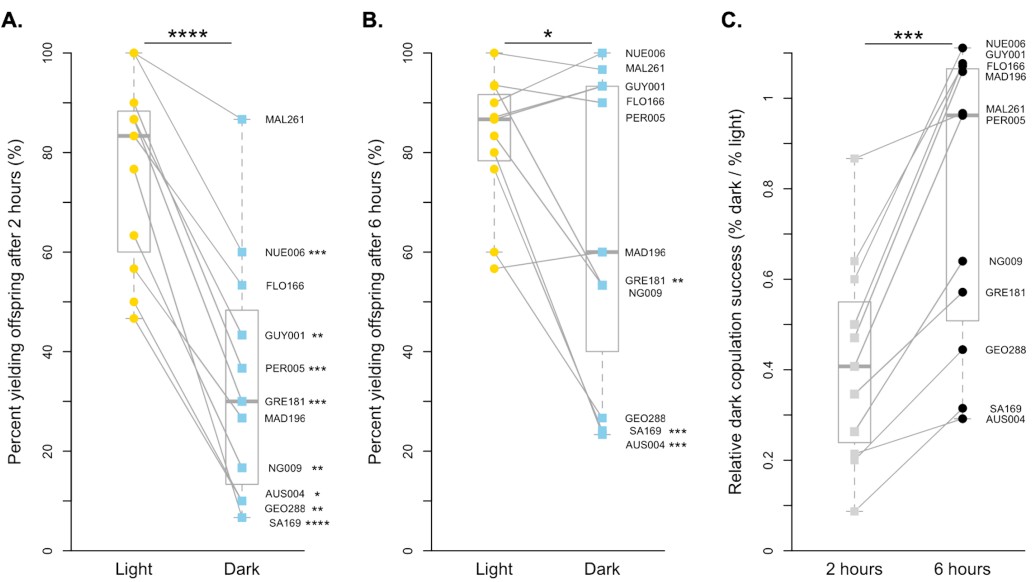

**Figure 1 Copulation success of 11 *D. simulans* strains by treatment and time point.** (A) The copulation success of pairs held for 2 h in the light (yellow circles) compared to in the dark (blue squares). (B) The copulation success of pairs held for 6 h in the light (yellow circles) compared to in the dark (blue squares). (C). The relative light independence of copulation success in the dark compared to in the light is shown for each strain for 2 h (light grey squares) and 6 h (black circles). For all, paired data points between treatments are connected with a line. Corresponding points are labelled with their strain label (Table 1). Individual strains and significance values after correction for multiple comparisons are indicated to the right of each point. Species wide differences are indicated with asterisks above plots ( * = $p < 0.05$, *** = $p < 0.001$, and **** = $p < 0.0001$).

a significant difference at the 6-hour exposure time (paired Wilcoxon $p = 0.5988$). The loss of the effect is driven by *D. simulans* strains improving their relative copulation success in the dark significantly when given more time (77.36%), compared to a more minor improvement by *D. melanogaster* (80.24%). We also found no correlation between the light-dependent copulation behavior of *D. simulans* strains and the *D. melanogaster* strains collected from the same (or similar) geographic location at 2 h (Pearson's $r = 0.1679$, $p = 0.6217$) or 6 h (Spearman's rho = 0.2115, $p = 0.5324$).

## Correlations of light-dependent copulation success

Because we found the largest effect of light-dependent copulation success at the 2-hour exposure time, we used the relative copulation success in the dark of our 11 *D. melanogaster* and *D. simulans* strains at 2 h to test for a correlation with other variables: aspects of collection location and date. For our *D. simulans* strains, we found no correlation between behavior and longitude (Pearson's $r = -0.2787$, $p = 1$, Fig. 3B), distance from the equator (Pearson's $r = -0.2579$, $p = 1$, Fig. S1A ), or collection date (Spearman's rho = 0.2152, $p = 1$, Fig. S2 A). For our *D. melanogaster* strains, we found no correlation between behavior and distance from the equator (Pearson's $r = 0.2490$, $p = 1$, Fig. S1B) or collection date (Pearson's $r = 0.0700$, $p = 1$, Fig. S2B ). We did, however, detect a significant correlation

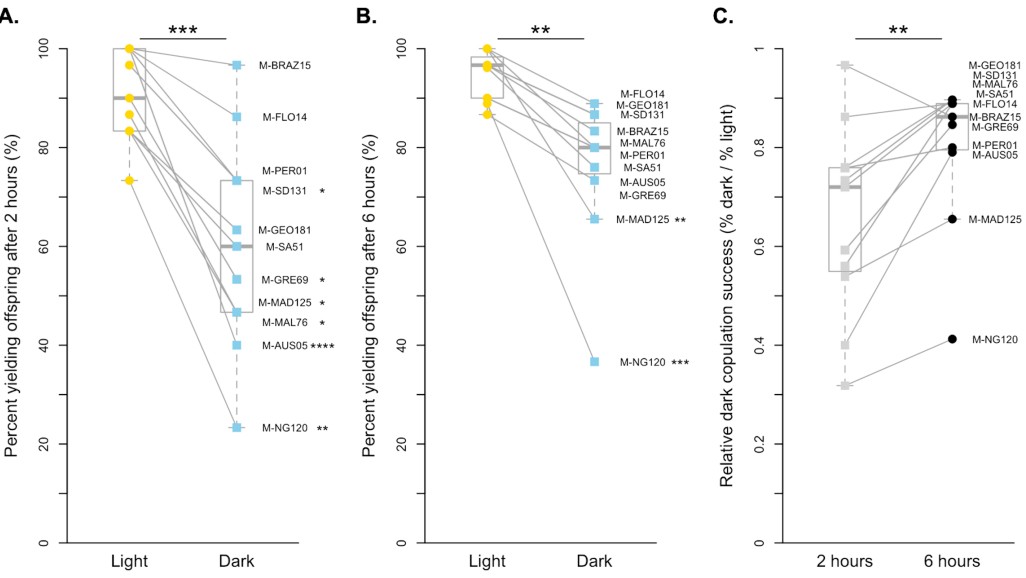

**Figure 2** **Copulation success of 11 *D. melanogaster* strains by treatment and time point.** (A) The copulation success of pairs held for 2 h in the light (yellow circles) compared to in the dark (blue squares). (B) The copulation success of pairs held for 6 h in the light (yellow circles) compared to in the dark (blue squares). (C). The relative light independence of copulation success in the dark compared to in the light is shown for each strain for 2 h (light grey squares) and 6 h (dark grey circles). For all, paired data points between treatments are connected with a line. Corresponding points are labelled with their strain label (Table 1). Individual strains and significance values after correction for multiple comparisons are indicated to the right of each point. Species wide differences are indicated with asterisks above plots ( ** = $p < 0.01$ and *** = $p < 0.001$).

between light-dependent copulation success at 2 h and longitude (Pearson's $r = -0.8617$, $p < 0.01$, Fig. 3C).

## Light-dependent courtship behavior

We wanted to know if light-dependent copulation behavior was mediated by male or female behavior. To test for differences in courtship behavior, we observed the courtship of two *D. melanogaster* strains under bright light and in darkness (Fig. 4A, Table S2A). For the strain we identified as relatively light-independent using our copulation assay, M-BRAZ15, we found that males court females at high frequencies in both treatments. Specifically, 100% of M-BRAZ15 males courted their own females under bright light, while 80% displayed courtship towards females when observed in the dark. Strain M-NG120, which displayed the most light-dependent copulation in our assay, courted females 50% of the time under bright light, and 30% of the time in the dark. While each strain showed a 20% decrease in overall courtship, the difference was not significant in either case ($p = 0.9474$ for both). Additionally, we detected no significant differences between courtship latency or effort for pairings observed in the light compared to those in the dark, however sample sizes are quite small, as not all males displayed courtship (Table S3).

We also observed the courtship behavior of two *D. simulans* strains under the same conditions (Fig. 4B, Table S2A). MAL261, which copulated successfully independent of

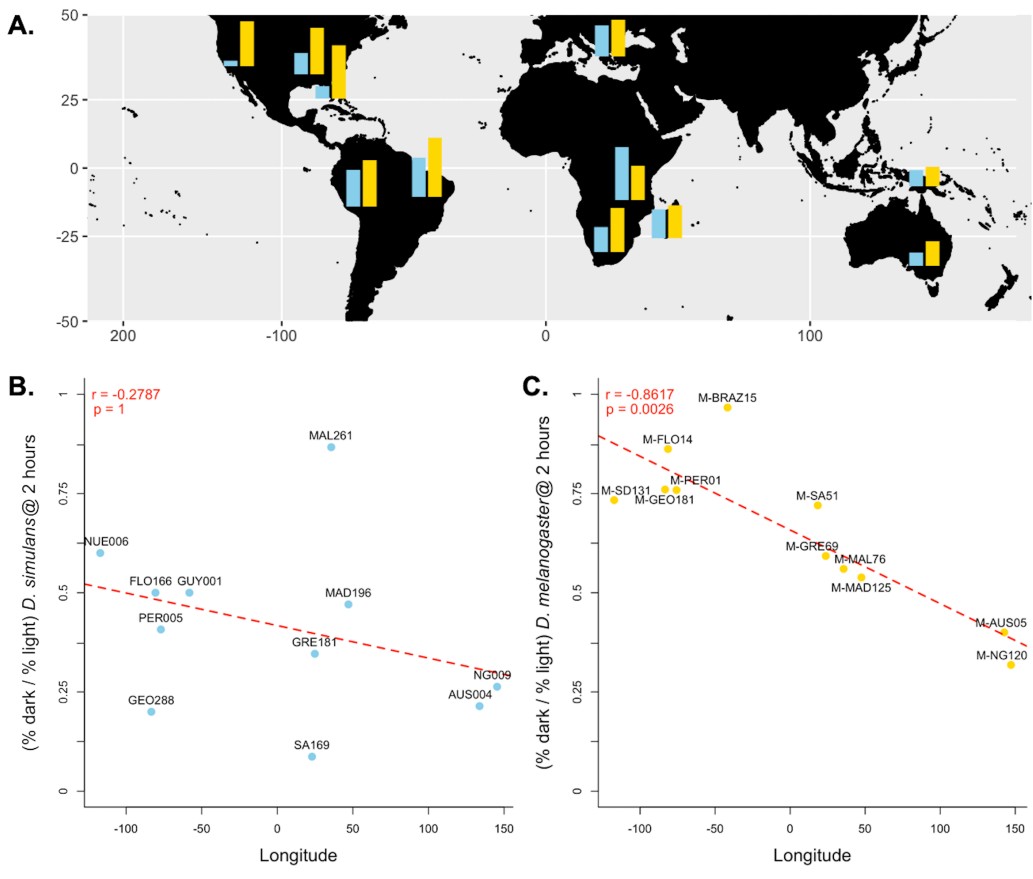

**Figure 3** **Differences in dark courtship ability correlate with longitude in *D. melanogaster* but not *D. simulans*.** (A) The relative dark copulation success at 2 h is plotted by geographic sample for *D. simulans* strains (blue) and *D. melanogaster* strain (yellow). (B) There is no correlation between relative light dependence at 2 h (*y*-axis) and longitude (*x*-axis) for *D. simulans* strains. (C) There is a significant correlation between relative light dependence at 2 h (*y*-axis) and longitude (*x*-axis) for *D. melanogaster* strains. For B and C, individual points are labelled with their strain label (Table 1). The red dashed line represents the best fit line from a linear model Pearson's correlation coefficient and significance values, corrected for multiple comparisons, are displayed in the upper left corner of the plot.

light, displayed high courtship towards their own females in both scenarios; 100% of males courted in the light, while 71.4% displayed courtship in the dark ($p = 0.4909$). For SA169 males, which showed the strongest signal of light-dependent copulation success in our mating trials, we observed a significant difference in the proportion of males that courted under bright light, 100%, compared to in the dark, 14.3% ($p < 0.05$). For both MAL261 and SA169 we did not detect significant differences in courtship latency or effort when courting in the dark relative to the light (Table S3). To identify whether the difference in the proportion of courtship we observed is driven by male or female behavior, we observed MAL261 males with SA169 females and SA169 males with MAL261 females under the same conditions (Fig. 4C, Table S2B). We found that 75% of MAL261 males still court SA169 females in the dark, compared to 66.7% in the light ($p = 1$). Interestingly, we also found that SA169 males court MAL261 equally as well in the dark as they do in the light (100%

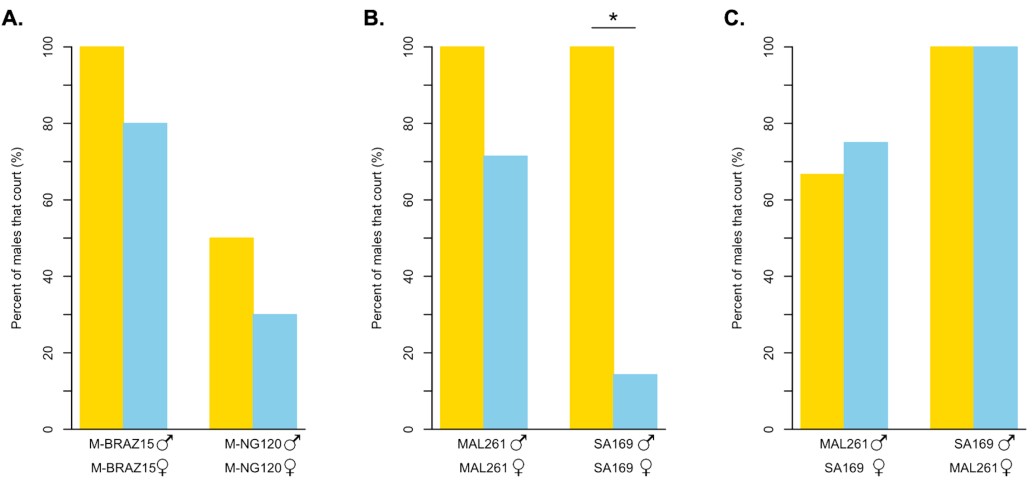

**Figure 4  Courtship behavior of light dependent and light independent *D. melanogaster* and *D. simulans* under bright light and in darkness.** (A) The percent of *D. melanogaster* strain males that court their own females in the light (yellow bars) compared to in darkness (blue bars). M-BRAZ15 was the most light-independent strain from our copulation assay, while M-NG120 was the most light-dependent ($N = 10$ for both). (B) The percent of *D. simulans* strain males that court their own females in the light (yellow bars) compared to in darkness (blue bars). $N = 4$ in the light, and $N = 7$ in the dark for both strains. (C) The percent of *D. simulans* strain males that court the opposing strain's females in the light (yellow bars) compared to in darkness (blue bars). $N = 3$ for all except for MAL261 males with SA169 females in the dark, where $N = 4$. For B and C, MAL261 was the most light-independent strain from our copulation assay, while SA169 was the most light-dependent. * $= p < 0.05$.

for both, $p = 1$). Again, there were no detectable differences in courtship latency or effort at these small sample sizes (Table S3).

## DISCUSSION

### A species difference in light-dependent copulation behavior

In some respects, our findings confirm a previously described species difference in light-dependent copulation behavior (*Grossfield, 1971*; *Spieth, 1974*). Specifically, *D. melanogaster* strains had, on average, greater copulation success in the dark than *D. simulans* strains when assayed for 2 h. However, this difference disappears when strains of each species are given 4 more hours. While both species show improvement in copulation success in the dark when given increased time, the loss of a species difference at 6 h is driven by a significant improvement in ability to copulate in the dark among *D. simulans* strains relative to *D. melanogaster* strains. Interestingly, the overall improvement is largely driven by 6 *D. simulans* strains (Fig. 1), that copulate equally as well in the dark as in the light when given 6 h.

In other measures, our results refute some of the species-wide conclusions made by previous studies of light-dependent copulation success, and address some of the inconsistencies among previously published results. First, we show that *D. melanogaster* strains, as a whole, do not copulate successfully independent of light. At both the 2 and 6-hour exposure times, we detect a significant difference in copulation success between

our light and dark treatments. Individually, in both treatments, each strain has a higher copulation success in the light compared to the dark (6 strains and 2 strains significantly so at 2 and 6 h respectively). This is in contrast to *D. simulans*, where a handful of strains show relatively unchanged, if not slightly increased, copulation success in the dark relative to the light at the 6-hour exposure time. Overall, while these data do demonstrate a difference between species in light-dependent copulation behavior, they also highlight an important role of both genetic variation and assay time in detecting this species difference. Our data show that a high level of intraspecific variation for both species creates largely overlapping distributions of behavior that are greatly affected by exposure time. Had we chosen fewer strains or sampled at a single exposure time, our conclusions may have been very different.

## Environmental correlates of light-dependent copulation behavior

We observed substantial variation in light-dependent copulation behavior among our *D. simulans* and *D. melanogaster* strains. Previous results have suggested a role of ecological generality in dark copulation success (*Grossfield, 1971*); species that occupy greater geographic range tend to have increased ability to copulate successfully in the dark. However, both *D. simulans* and *D. melanogaster* are near cosmopolitan human commensal species that overlap nearly entirely in their geographic ranges (*Kliman et al., 2000*). To attempt to identify other potential causes of the species difference we detected at the 2-hour exposure time, we tested for correlations between light-dependent copulation success and other aspects of strain collection (latitude, longitude, and collection date). We found no correlation between collection date and behavior for either *D. melanogaster* or *D. simulans*, which reduces (but does not conclusively eliminate) the likelihood that variation in light-dependent copulation behavior is a product of laboratory adaptation. For *D. simulans*, light-dependent copulation behavior did not correlate with either longitude or distance from the equator. Further studies including more strains would have increased power to detect less significant correlations, however. For *D. melanogaster*, behavioral variation correlated strongly with longitude at the strain's collection site (Fig. 3C). This correlation is unlikely to be a result of differences in habitat, as longitude does not correlate strongly with measures of environment. Instead, this correlation might reflect the demographic history of *D. melanogaster*. If so, the lack of correlation between variation in *D. simulans* behavior and longitude potentially reflects differences in species demography.

   *D. melanogaster* originated in sub-Saharan Africa, eventually expanding out of Africa and colonizing the rest of the world, first colonizing Eurasia (*Li & Stephan, 2006*). Much later, an admixed American population was established, likely during the modern colonization of the Americas (<500 years ago) (*Duchen et al., 2013*). Australia's population is similarly admixed and likely very recently colonized with modern sea travel (*Arguello et al., 2019*). Importantly, there is still significant gene flow between populations (*Arguello et al., 2019*). Our data potentially mirror these two recent independent trans-oceanic colonization events. American populations show the highest level of light-independent copulation success, while Australian/south east Asian populations display the lowest. The African and European populations fall in the middle (for both behavior and longitude). If we consider the African phenotypes ancestral, then there appears to be little change with the colonization

of Europe, but dramatic shifts in behavior with opposing valence for the colonization of the Americas and Oceania. Whether these differences are due to founder effects or divergent selection remains an open question. In contrast, *D. simulans*, which originated in East Africa or Madagascar, spread across the globe much more recently (*Dean & Ballard, 2004*), and globally distributed samples show significantly less population structure (*Irvin et al., 1998*) and clinal variation (*Machado et al., 2016*) than *D. melanogaster*. Similarly, there is no discernable structure to light-dependent copulation success in our data. The effects of species demography on light-dependent copulation success are hypotheses that still require explicit testing, however. We cannot make strong conclusions regarding these effects with the data we have presented here.

## Courtship behavior differs between relatively light-dependent and light-independent strains

To begin to identify the mechanistic drivers of differences in light-dependent copulation behavior, we selected the most light-dependent and light-independent strains from each species to compare male courtship rates in both the light and the dark. For *D. melanogaster*, we found that both the light-independent (M-BRAZ15) and light-dependent (MNG120) strains courted at statistically indistinguishable rates in both treatments. Although, M-NG120 showed reduced courtship overall. This relative reduction in courtship is congruent with the result of our mating assay, where M-NG120 showed the lowest copulation success in both the light and the dark, indicating this pattern is more likely a result of differences in male courtship vigor. These results imply that for these strains of *D. melanogaster*, reduced copulation success may depend partially on a male's ability to locate females in the dark. Partly supporting this hypothesis, recent work has shown that other *Drosophila* species display increased courtship latency in the absence of visual cues (*Roy & Gleason, 2019*). We did not detect significant differences in courtship latency, but further observation at larger sample sizes may find a similar trend. Differences in ability to locate females in the dark may be driven by a unique male scanning behavior described among dark-courting *D. melanogaster* strains, presumably used to locate females without visual input (*Krstic, Boll & Noll, 2009*). *D. melanogaster* males also depend on olfactory cues and female movement (and the resulting sound/vibration) to initiate courtship with females in the dark (*Ejima & Griffith, 2008*; *Stockinger et al., 2005*). Thus, in the absence of visual detection of movement, *D. melanogaster* males can rely on another sensory modalities to identify the presence of courtship targets and adopt a different strategy to locate them in the dark. The variation we observed among strains also potentially reflects variations in male's ability to locate females, or variation in female signals that males use to locate females in the dark (*Trajković et al., 2017*).

For *D. simulans*, we found that the light-independent strain (MAL261) courted females with high frequency in the light, and somewhat reduced, although statistically insignificantly, frequency in the dark. For the light-dependent *D. simulans* strain (SA169), we found that males courted females at high frequencies in the light, but at significantly lower frequencies in the dark (Fig. 4B). Thus, SA169 males are less willing or able to court SA169 females in the dark than MAL261 males are able to court MAL261 females,

despite having equal courtship in the light. From these results, it is unclear if the difference in the amount of courtship in the dark is driven by males or females. It is possible that SA169 males cannot identify potential courtship targets because of the lack of a perceivable moving courtship target. It could be equally likely that SA169 female signals are missing or become indiscernible in the dark, leading to a reduction in male courtship.

To test if the loss of SA169 male courtship is driven by female signals, we swapped female types for our light-dependent and light-independent strains. We observed SA169 males with MAL261 females, and vice versa. MAL261 males were equally able to court SA169 females in the light and in the dark. Surprisingly, we found the same pattern for SA169 males: they court MAL261 females 100% of the time in both the light and in the dark. So, SA169 males can locate and court females in the dark, but do not do so when those females are also SA169. In contrast, MAL261 males will court either female in the dark. These results indicate that there is both a difference in male strains' ability/willingness to court and in female strains' attractiveness in the dark. It is possible that these differences are the result of differences in female activity or male olfactory or vibratory/acoustic perception ability in the dark, but we do not know if *D. simulans* males rely on olfactory, auditory, or vibratory cues to identify females in the dark in the same way that *D. melanogaster* males do (*Ejima & Griffith, 2008*). Ultimately, these results highlight the complex coordination of signals and receivers that underlies *Drosophila* courtship. They also suggest that the variation we observe among lines, potentially in both species, can reflect variation in male behavior, female behavior, or both. More careful observation of a greater number of flies and strains (and strain combinations) is necessary to understand the contributions of males and females.

## The role of sensory perception in *D. simulans* light-independent courtship behavior

The above results highlight an important, yet unknown aspect of male mate choice in *D. simulans*. While we know quite a bit about the signals that males send to females during courtship (*Greenspan & Ferveur, 2000*), it remains unclear what signals *D. simulans* males use to discriminate between males and females. Unlike its sister species, *D. melanogaster*, *D. simulans* males and females express the same primary chemotactic pheromone, 7-tricosene (7T) (*Cobb & Jallon, 1990*). It is possible that males discriminate between sexes using a difference in 7T abundance, a difference in one of the many low-abundance cuticular hydrocarbons (CHCs) (*Pardy et al., 2019*), or the male-specific expression of cis-vaccenyl acetate (*Jallon, 1984*). While *D. simulans* males and females can differ quantitatively in CHCs, this only seems to be the case in some strains (*Sharma et al., 2012*). Additionally, the perception of these chemotactic signals is light-independent, and are unlikely to explain the species-wide reduction in dark copulation success anyway. Our results suggest that male courtship initiation in *D. simulans* may partially depend on a female visual signal. They also suggest that strains can vary in the importance of visual perception to initiate male courtship, which might reflect variation in ability to rely on other sensory modalities to identify female courtship targets.

## CONCLUSIONS

The above conclusions highlight two potential areas of further investigation. First, for males that show higher light-dependent copulation success, what are the visual signals necessary for courtship initiation? Second, for males that show relatively light-independent copulation success, what other signals allow them to successfully initiate courtship in the dark? In either case, the role of vision needs to be further examined in combination with the other senses in a larger comparative framework to understand how sensory modalities interact to determine variation male courtship behavior both within and between species. Should visual signals prove important to *D. simulans* male mate discrimination, there is ample opportunity for a more precise characterization of the neural substrate of signal hierarchies in *D. simulans* courtship behavior. In *D. melanogaster*, sensory receptors (*Ahmed et al., 2019*; *Ejima & Griffith, 2008*; *Göpfert & Robert, 2003*; *Montell, 2009*) and specific neurons necessary to detect a variety of courtship signals (*Lu et al., 2014*; *Pan, Meissner & Baker, 2012*; *Seeholzer et al., 2018*; *Starostina et al., 2012*; *Thistle et al., 2012*; *Toda, Zhao & Dickson, 2012*) and produce male courtship behavior (*Kohatsu, Koganezawa & Yamamoto, 2011*; *Von Philipsborn et al., 2011*) have been identified. Our results highlight that *D. simulans* may rely on different (or differentially weighted) female signals. A comparative analysis of the molecular underpinnings of signal perception in these species will help to identify differences in signal perception or signal hierarchies.

## ACKNOWLEDGEMENTS

We are grateful to Dr. William R. Rice for the casual laboratory observation that inspired this project.

### Funding

This work was supported by the NIH (R01 GM098614). The funders had no role in study design, data collection and analysis, decision to publish, or preparation of the manuscript.

### Grant Disclosures

The following grant information was disclosed by the authors:
NIH: R01 GM098614.

### Competing Interests

The authors declare there are no competing interests.

### Author Contributions

- Michael P. Shahandeh conceived and designed the experiments, performed the experiments, analyzed the data, prepared figures and/or tables, authored or reviewed drafts of the paper, and approved the final draft.
- Cameryn Brock conceived and designed the experiments, performed the experiments, analyzed the data, authored or reviewed drafts of the paper, and approved the final draft.

- Thomas L Turner conceived and designed the experiments, authored or reviewed drafts of the paper, and approved the final draft.

## Data Availability

The raw measurements are available in the Supplementary Files.

## Supplemental Information

Supplemental information for this article can be found online at http://dx.doi.org/10.7717/peerj.9499#supplemental-information.

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
