# Peer review of "Light dependent courtship behavior in Drosophila simulans and D. melanogaster"

_PeerJ, doi:10.7717/peerj.9499_

## Round 0.1 · original submission · Major Revisions

I highly recommend you to send a new version taking into account the of reviewers comments. The reviewers have valued the work and agree to review the new version.

Reviewer 1 ·

Basic reporting

I’m not native English spiker but I think that the language is good enough, it is clear and fluent.
The introduction is generally well written. Specific comments are provided in the pdf file.
Figures match the criteria and raw data was provided.

Experimental design

The paper covers the material well known to the scientific community. However, the authors believe that the previous researches do not cover a large enough sample to represent a whole species. In my opinion, the sample size in this research is small too. If the authors aimed to increase the sample size, in addition to increasing the number of strains, the number of pairs needed to be increased too. The 1320 pairs are still not enough to represent the whole species. It is necessary to either change the approach, not emphasize the sample size, or increase the number of pairs. Authors could switch the focus to the correlation between light-dependent copulation success and collection variables (longitude and latitude).
Why singing and copulation attempts have not been analyzed and discussed? What is the connection with this work?
Specific comments are provided in the pdf file.

Validity of the findings

Underlying data have been provided. Statistical analysis is appropriate. The conclusions are clearly defined.

Additional comments

Generally, the paper is interesting well and writhen. However, it is necessary not to underline the sample sizes. There are more interesting findings. One of them is the correlation between light-dependent copulation success and demographic history of species.

Annotated reviews are not available for download in order to protect the identity of reviewers who chose to remain anonymous.

Reviewer 2 ·

Basic reporting

The ms is clearly written except the second part of the discussion which is vague. The literature is mostly relevant but several critical references are missing. The illustrations are OK except for a color code detail. The raw data are supplied.

Experimental design

This ms presents relatively original data given that a couple of similar papers appeared in the late 1970’s (Joe Grossfield) also testing the inter- and intraspecific effect of light on Drosophila mating. The research question is well defined. The investigation was performed with great rigor. The methods are described with enough details to allow replication.

Validity of the findings

The novelty is not completely true (see my comment above). The data are statistically sound but there is some misinterpretation related to the longitude parameter. The speculation is questionnable. Some of the conclusions are well stated, but not completely supported by the data shown. Some supplementary data should be added to provide a clear answer and to support more firmly the current conclusion.

Additional comments

Paper by Shahandeh and coll

This paper deals with the effect of light on a set of populations of two sibling Drosophila species: D.melanogaster and D.simulans. These populations, obtained from stock centers, were initiated by flies collected at different years. The authors have tried to compare pairs of populations (in the two species) collected nearby. The principal experiment consisted to indirectly measure mating frequency (based on fecundity) between pairs of flies either tested under white light or under darkness, during 2 hours and during 6 hours. Based on the ratio between “mating” frequency under the two light regimes (only for 2-hour long tests) populations were classified based on their ratio (between mating frequencies under light/darkness) during two hours. Authors have tried to establish correlations between this ratio and various geographical parameters (longitude, latitude). They also measured the “real” courtship response in the two extreme “ratio” lines in each species under each light regime and they performed a reciprocal cross experiment between two extreme ratio D.simulans lines.

I found generally the experimental design of the paper solid, but the most intersting findings were not developed enough to convey the authors to a strong conclusion. Moreover, a large part of the Discussion is too vague and should be completely rewritten.

Major points:
A better description of the Drosophila populations tested should be provided into a complete table showing not only the coordinates but also the altitude at which the founder flies were collected, together with the type of vegetation and climate.

LL 225-236 : It would be worth to complete this paper with cross experiments with the two extreme D.melanogaster lines (similarly to what was done with D.simulans).

The Discussion is loose and too vageu, specially in its second half.

Minor points:
L 43: the reference “Savarit et al., PNAS 1999” needs to be added

L 95-96: Keeping males together until the mating test can strongly affect their wiliingness to mate depending on male genotype (Svetec et al., Genetical Research Camb, 2005). Please explain why you have kept together before the test.

L102: The presence of food in the test can interact with the presence/absence of light as shown with two D.simulans populations and two D.melanogaster populations (Cobb & Ferveur, Behavioural Processes; Table 1). Please discuss.

L158/L177: “...more successful...” than what?

LL 206-207: The longitude scale is not a natural parameter per se given that it is only based on the convention of the 0° at Greenwitch meridian. If, for D.simulans (Figure 4B), one shifts the world map representation starting from left to right in Oceania (100-150-with AUS004 + NG009 lines) then continuing with Pacific ocean and America and ending with Africa/Europa, I would be curious to see the level of the correlation?

I generally find that the longitude is not a real biological parameters, and I think that oceans and deserts are more likely to cause separation between populations. To me, the latitude would represent a more potent factor to investigate, despite the fact that the number of populations investigated here is too limited to establish a real strong correlation. Moreover, the pairs between sibling species should be examined more carefully than just belonging to same country (another artificial non-biological factor) but should rather be related to ecological and geographical parameters such as altitude relatively to the sea level, type of vegetation (forest vs desert), seasonal variation of temperature etc...


L235 : delete « interestingly ».

LL272-273 : I found completely non-realistic to test the effect of collection date given that it can be confounded with so many other parameters examined, or not examined here (longitude, latitude, light x food interaction, group effect...). Otherwise, the authors could always carry a study similar to that performed on a smaller geographical range— Houot et al. J.Exp. Biol. 2010—to test the effect of acclimation on mating and pheromones after several years spent in the laboratory).

L276-280 : I do not agree with this conclusion made and with the explicative scenario provided (LL281-286).

L314 « absences » or « absence » ?

L317 : One cannot conclude this since complementary tests are missing (see above and below).

LL333-334 : I found that the cross population mating experiment probably provides the most exciting data ofthe paper. To validate this result, the authors should test another cross (reciprocal crosses between MAL261 x GEO288 or MAL261 x GRE181).

L337 : « ...possible...are... »

L351-352 : 7-T is not very different between D.simulans sexes, but only males produce cis-vaccenyl acetate (Jallon, Behav Genetics 1984).

L357 : « ...suggest that male latency to initiate courtship... »

L367 : « « ...what other signals cause... » please rewrite.

LL367-377 : the style is loose : the authors should rewrite (not « we’ve » !)

L373 : « we’ve » what does this mean ? It seems that « other authors have identified... »

L369 : « smell-blind males in the dark ? Tarsi-less males ? ». The authors should refer to published papers: Robertson Experiencia 1983; Grillet et al., R.Soc. Open Sci. 2018.

LL377-378 : I feel that the authors have not exploited the most interesting point of their study: instead of inter-specific difference, they should have explored more deeply inter-populations (intraspecific) variation. This is specially worth when looking at the data shown on the Figures 5B & C (see also my point above).

Reviewer 3 ·

Basic reporting

This paper addresses the use of light (and by inference, vision) in courtship behavior in Drosophila melanogaster and D. simulans. Although mating in the dark has been explored extensively in these species (much more so in D. melanogaster than D. simulans), what is unique in this paper is the use of multiple strains from different longitudes. This is probably the best assessment of species variation in this trait that I have seen (if not the only one). The raw data are solid, and most is included (the data for longitude, latitude, or age of strain are missing), but the analyses (problems detailed below) seems to have many problems.

In general, the discussion of visual assays, except within a neurobiology context, is very dated. Many people have done more recent assessment of visual dependency using other approaches and assays (such as painting over eyes, e.g. Giglio & Dyer 2013, Ecology and Evolution 3: 365).

Experimental design

For the most part the experimental design is good, but the sample sizes are too small for the light/dark observations (N=10 for D. melanogaster and N=3-7 for D. simulans without the same numbers done for comparisons). I didn’t find reports of the sample sizes in the text—they are in Table S2. I do not think the sample sizes are sufficient for strong conclusions to be drawn from the data. Moreover, with careful observation in the dark, many of the questions about the visual cues necessary for males to start courtship can be addressed. The authors missed the opportunity to calculate courtship latency in these assays (they probably have the data), which might provide clues about how males court without vision. A recent paper (Roy and Gleason, 2019, Behavioural Processes 158:89) measured courtship parameters in the dark in other melanogaster group species providing evidence that males of some species start courtship in the dark more slowly than in the light (but some are equally fast). Such analyses might provide insights into the differences between D. melanogaster and D. simulans.

There are a few areas where the materials and methods are not clear. These are below with the line comments.

Validity of the findings

I have many questions about the data analysis, particularly the calculation of Pdark/Plight and the way it is graphed. Looking at the data in Table S1, some of the these values are negative, which is not possible with this data. I recalculated the values and mine match what is in the table for the 2 hour D. simulans, but after that the numbers are incorrect. The values are graphed in Figure 2C. First, I don’t understand why the y-axis has a scale of 0-110 and not 0-1.1 (the correct scalce for a percentage divided by another percentage scale). Secondly, if there a negative values, these should appear on the figure. Thirdly, the numbers are clearly not correct in the figure, both based on the numbers in Table S1 and the values in Figure 2A and Figure 2B. For example, after 6 hours, NUE006 mated more in the dark than the light, thus Pdark/Plight should be greater than 1. This strain appears in Figure 2C nearly at the bottom of the figure, where as it should be closer to the top. Thus all the analyses are questionable.

This is also a problem in Figure S1A. Only three of the points are in right place on the Y axis (if assuming the scale has been multiplied by 100). Thus, I don’t think the analyses done with longitude, latitude, or age of strain are correct (and the data for these analyses are not included, but they should be). I think it is possible that the authors transformed the data, but this is not explained (and also does not explain why three points are correct and the rest are not).

At this point I stopped evaluating the paper for the results based on the 2-hour and 6-hour data because I think all the analyses have to be repeated.

I was also looking for a discussion of the biological relevance of doing 2 and 6 hour assays. I was expecting a discussion of how short these observations are with respect to many that Grossfield and others have done. In addition, I was expecting an explanation for changing to an only 30 minute assay for the observation tests.

Additional comments

Line 50: use of the word “surprising.” Although it is true that many species have some light dependency, there are also a large number that are light independent. Basing D. melanogaster behavior on that of other species does not seem to be a great argument.

Line 55: Please explain what a “postural display” is. I am unfamiliar with this.

Line 61: Please avoid the use of contractions. Also, the males in this experiment were not physically blinded, but were mutant for a particular gene. The point that when males cannot see they do not do as well as wt males is valid, but make it clear that this is a genetic mutation. To further support this, mutant white males, which are blind, also do not court as well as wt males.

Line 65: “assays of variable lengths and designs.” It would be helpful to the reader to understand what are the different past approaches to these kinds of experiments. In addition, the citations here, from 1959 and 1950, predate even Grossfield’s (1971) approach. Moreover, many people have studied light dependency more recently, thus variability in studies should be updated.

Line 93: How close to eclosion were virgins collected?

Line 101: How long after lights on was the experiment performed?

Line 103: Given that this is a paper about vision, a description of the light used would be helpful (For example, wavelengths, light source, Lumosity, distance from vials—all these things may not be necessary but provide at least a description of the light bulb).

Line 111 (and throughout the paper): The two measures are not different timepoints, but rather different exposure times. When I read the abstract I assumed that different timepoints were used such as 2 hours post lights on and 12 hours post lights on.

Line 125: use of collection date: not all strains in Table 1 have collection dates.

Lines 130-142: Were the dark and light experiments performed on the same day?

Line 134: How long after lights on was the experiment performed?

Line 136-139: I found this description confusing. I think what happened is that for each minute, the presence/absence of the three courtship behaviors was scored. Rephrase to make that clear. In addition, why collect all this data when what is reported is only whether or not the males courted. How was it decided that males did or did not court? There is a lot that can be done with the data here, such as calculating courtship and copulation latency (do they differ in the light and dark).

Line 141: Dark room with red light: make it clear that the sole illumination in the room was red light.

Line 143: What timepoint was used to determine “most” light independent and light dependent?

Line 147: Please identify the strain

Line 150 : “their own females” – slightly confusing given that one strain did not have the same females.

Results: throughout, please specify the test used when reporting P values.

Line 159: The value I obtained here was 0.45 (Pdark/Plight)

Line 164: The value I obtained here was 0.78 (Pdark/Plight)

Line 267: Replace “space” with “range.”

Lines 361-370: a list of questions are hard to read. You have a single point: vision needs to be combined with other sensory modalities for analysis to determine how the senses interact.

Line 373: avoid contractions. Also, the use of “we” is misleading here.

Author contributions: the contribution of one author is not mentioned.

Table 1: Organizing this table by collection location would help the reader considerably.

Figure 1: the experimental design is simple enough that I don’t think the figure is necessary.

Figure 5: Because D. simulans results are always presented first in the rest of the paper, present them first here.

Figure S1: An indication of units on the X-axis would be helpful. The label on the Y-axis is confusing.

Table S1: the Fisher’s corrected P-values are missing for all the D. melanogaster comparisons. The last cell that is occupied as an incomplete formula and not any values.

---

## Round 0.2 · Major Revisions

The reviewers have revised the manuscript for a second round. They have done a great job and devoting quite a time to improve this manuscript. Reviewer 2 and 3 are not fully satisfied with this version. There are still some aspects that need to be addressed. I do also think there are a lot more room for more details and justification in the methodology (e.g. about keeping males together until the mating test and whether males copulate in the observational period). There is a need for more increase consistence as well (e.g. text and figure). Several missing information was pointed out for reviewer 2. I would like to remind authors that PeerJ requires that methods should be described with sufficient information to be reproducible by another investigator. I expect all these aspects fully covered in a new and hopefully satisfying version.

Reviewer 1 ·

Basic reporting

The authors did their best to respond to all requests of the reviewer. The paper is now acceptable for publication in PeerJ.

Experimental design

The authors justified the experimental design.

Validity of the findings

Underlying data have been provided. Statistical analysis is appropriate. The conclusions are clearly defined.

Additional comments

The paper is interesting, well writhen and acceptable for publishing.

Reviewer 2 ·

Basic reporting

Clear language.

Experimental design

Questionnable specially after reading their reply to my requests.
Primary research not original and not so rigorous.

Validity of the findings

The findings miss some key experiments to allow any interpretation.
Conclusion remain vague.

Additional comments

I am sorry to say that I am not satisfied with the way they dealt with my request. While they took into account most of my minor remarks (concerning the typo and grammatical problems), they did not answer, or they negatively answered to my requests. I just give soma exemple below.

To my question: Keeping males together until the mating test can strongly affect their willingness to mate depending on male genotype (Svetec et al., Genetical Research Camb, 2005), the authors replied: The study the reviewer provides compares white-eyed flies harboring a known mutation that affects male sex discrimination behavior during courtship to wild-type strains. It is also known that the white mutation significantly impairs vision which we know will also impact courtship. It would be surprising if the authors didn’t find an effect of genotype here. In fact, it is is not true since wild-type red eyes males were also tested in this paper (Cs, Di2, Cs/B42 and Di2/B42; Table 2) and showed a moderate to strong effect either in homo- or heterosexual courtship (or in both) between isolated vs grouped condition.

To my question: "The longitude scale is not a natural parameter per se given that it is only based on the convention of the 0° at Greenwitch meridian", the authors replied: "The convention of 0 at the Greenwich meridian does however place 0 very close to the source populations (sub-Saharan Africa and Madagascar) for both of these species, so it is not entirely without meaning in this case". So does this reply bring any fuel to my question?

Reviewer 3 ·

Basic reporting

In the text, the change in mating from light to dark is discussed as a percentage, but in Figure 1C and 2C it is graphed as a ratio. I prefer the ratio so that the point being discussed is not confused with the data points at each time and condition. Regardless, the text and figure should be consistent with each other to help the reader.

Although I like the addition of courtship latency and courtship effort, I would like some more details about how these were calculated. First, did any of the males copulate in the observation period. If so, this should be mentioned, particularly because it affects the calculation of courtship effort (which the time spent in active courtship divided by the time from the initiation of courtship to copulation, not to the end of the observation period). The calculation of courtship effort (often called “courtship index”) is not clear.

In addition, the sample sizes in Table 2, I believe, are for the total number of pairs observed. If this is for the number of pairs in which the males copulated, then the sample sizes for Figure 4 are unknown. I think these sample sizes are actually the right sample sizes for Figure 4 (which should have sample sizes included in the legend or the figure) and the sample sizes here are even smaller if only the courting males are included (Line 170). This is then problematic because only ~15% of SA169xSA169 males courted in the dark. With a sample size of 7 (Table 2), this is equivalent to 1 male courting and the values in Table 2 have a standard deviation, which is not possible. I’m not sure where things are disconnected, but I’d like to see some clarity on sample sizes.

P-values were corrected for multiple tests but never is it stated for how many tests each P-value was corrected. I’d like to see that stated explicitly. The uncorrected P-values should also be reported in Table S1

Experimental design

The experimental design seems mostly solid (with the exception of some of the analysis described above). The geographic and date analysis is not explained in the materials and methods.

Validity of the findings

I do not find any problems with the findings. Quite a bit of it is speculation, but I do not feel it goes beyond the data available anymore.

Additional comments

Overall, this is a nice revision of the paper. The authors handled the criticisms well and I’m glad to see that the changes necessary in calculating the data did not change the conclusions. I think this paper will be noted by people interested in variation in courtship behavior.

Some specific comments:

Paragraph starting Line 47: This paragraph is difficult to read because it shifts ideas so many times. Breaking it up and concentrating on the message would help.

Line 53: Use of the word “surprising” not warranted. D. melanogaster uses many sensory modalities in courtship and has never been demonstrated to depend on one solely. Most tests of success in light and dark are measures of copulation success and not measures of courtship ability. This paper, as well as others recently, shift the emphasis on short term effects of vision from the long-term effects that have been tested before. Thus, “surprising” isn’t the right word when the previous analyses have been so different.

Line 221: “counterparts”: I found this confusing. Did you mean that you compared a D. simulans strain with the D. melanogaster from the same geographic location? I think this could be stated more clearly.

All figures: the font is so small that they are very hard to read.

Figures 1 and 2: yielding is not spelled correctly (y-axis)

Figure 1C and 2C: the contrast is hard to see (light gray and dark gray)

Table 2: what are the units for courtship latency?

Table 2 figure legend: “Mean courtship latency is prevented with standard deviation in parentheses. I think “presented” is meant instead of “prevented.”

---

## Round 0.3 · Minor Revisions

Now we have the review report back from R3, I encourage to send us a new version considering their comments.

Reviewer 3 ·

Basic reporting

The figures are well presented and the raw data is provided. The writing is mostly clear.

Experimental design

The experiments are straight forward and the revision makes the procedures clear and easy to understand.

Validity of the findings

There are problems with the conclusions because many of the conclusions are unsubstantiated.

The conclusion that there is variation within species for copulation success in the dark is robust, and should be of interest to people studying these species. The result that copulation success in the dark varies with longitude in melanogaster is surprising, but seems genuine.

However, the sample sizes for the observations are so small that no conclusions can be drawn. This is an unfortunate aspect of Drosophila behavior that the flies do not always behave when they should. The only aspect of this with nearly a reasonable sample size is the comparison between courting/non courting in the light/dark. With so few males courting, nothing can be concluded for courtship latency and courtship effort. The data could be reported in a supplementary but should be left as inconclusive. The point is that males might court less in the dark but it may be dependent on the courtship target, implying that failure to mate in darkness lies with the females. Nonetheless, is there power in the sample size to detect a difference? Whether males or females are driving reduction in mating remains an open question. The interpretation of these data needs to be reduced and adjusted to fit what the study can test, which with such a small sample size, is very limited. Some specific areas that need adjusting are below, but overall, the discussion of these results should be considerably shortened.

Additional comments

Line 127-128: Instead of “approximately 30 pairings,” give the range so that the reader knows that the sample size is sufficient (I had to go to the supplementary table to check this—make it obvious).

In Figure 4, the melanogaster results are presented before the simulans results, where in all the rest of the paper simulans comes first. The reader would be helped if these were reversed (Table 2 is set up with simulans first and it is the same data).

Line 263 (and elsewhere in the paragraph): Please do not claim that there was an increase if it is not statistically significant.

Line 376: “…SA169 is less willing or able to court in the dark than MAL26…” This is clearly not true if the courtship target is changed (Figure 4C). Please change the statement.

Line 385: “While we observed a reduction in MAL261 courtship overall when paired with SA169 females…” Where are the statistics that support this claim? The experiments were not performed at the same time (intra- and inter- strain courtship) so it is not possible to determine if the effect is real (if significant) or an artifact of performing the experiments at different times.

Line 418. “It is also clear that strains vary in the importance of visual perception to initiate male courtship success.” This is not clear. Only two lines were tested and male courtship initiation seemed to vary with courtship target more than with visual signals and cues.

---

## Round 0.4 · accepted · Accept

The new version fully addressed comments made by the reviewers. I think this will contribute to the knowledge in the area. Congratulations again on this. Good job.